# Changes in Dynamic Strength Index in Response to Strength Training

**DOI:** 10.3390/sports6040176

**Published:** 2018-12-19

**Authors:** Paul Comfort, Christopher Thomas, Thomas Dos’Santos, Timothy J. Suchomel, Paul A. Jones, John J. McMahon

**Affiliations:** 1Directorate of Sport, Exercise and Physiotherapy, University of Salford, Salford, Greater Manchester M6 6PU, UK; t.dossantos@edu.salford.ac.uk (T.D.); timothy.suchomel@gmail.com (T.J.S.); p.a.jones@salford.ac.uk (P.A.J.); j.j.mcmahon@salford.ac.uk (J.J.M.); 2Football Performance and Science Department, Aspire Academy, Doha, Qatar; c.thomas2@edu.salford.ac.uk; 3Department of Human Movement Sciences, Carroll University, Waukesha, WI 53186, USA

**Keywords:** isometric mid-thigh pull, countermovement jump, force, jump height, time to take-off

## Abstract

The primary aim of this investigation was to determine the effects of a four-week period of in-season strength training on the dynamic strength index (DSI). Pre and post a four-week period of strength-based training, twenty-four collegiate athletes (age = 19.9 ± 1.3 years; height = 1.70 ± 0.11 m; weight 68.1 ± 11.8 kg) performed three isometric mid-thigh pulls and countermovement jumps to permit the calculation of DSI. T-tests and Cohen’s effect sizes revealed a significant but small (*p* = 0.009, *d* = 0.50) decrease in DSI post-training (0.71 ± 0.13 N·N^−1^) compared to pre-training (0.65 ± 0.11 N·N^−1^); however, when divided into high and low DSI groups, differential responses were clear. The low DSI group exhibited no significant or meaningful (*p* = 1.000, *d* = 0.00) change in DSI pre to post-training (0.56 ± 0.05 N·N^−1^, 0.56 ± 0.09 N·N^−1^, respectively), whereas the high DSI group demonstrated a significant and large decrease (*p* = 0.034, *d* = 1.29) in DSI pre to post-training (0.85 ± 0.05 N·N^−1^, 0.74 ± 0.11 N·N^−1^, respectively), resulting in a significant and moderate difference (*p* = 0.034, *d* = 1.29) in the change in DSI between groups. These results demonstrate that DSI decreases in response to strength training, as expected, due to an increase in isometric mid-thigh pull peak force, with minimal change in dynamic (countermovement jump) peak force.

## 1. Introduction

To ensure that the previous phase of training has resulted in the desired changes in performance and to identify priorities for subsequent phases of training, accurate monitoring of changes in performance in response to training is essential [1]. As such, monitoring changes in countermovement jump (CMJ) performance (using force plate assessment) is commonly implemented in applied situations, likely due to its efficiency and utility [2,3,4]. Appropriate analysis of CMJ force-time data can provide detailed insight not only into changes in jump height but also jump strategy, with time to take-off (TTT) (sometimes referred to as contraction time) being indicative of changes not only in movement time but also countermovement displacement (squat depth) [5,6,7]. This level of detail is advantageous, as it is common to see no change in jump height but a change in jump strategy in response to training and competition [8].

While assessment of force production during dynamic tasks, such as the CMJ, is efficient and informative, the highest propulsive force, or peak force (PF), achieved during such a task is not indicative of maximal force production capabilities as these will be influenced by the athlete’s jump strategy. It is important, therefore, to include a method of assessing maximal force generating capacity to determine if the athlete should emphasize rapid force production or maximal force production during the subsequent training phase. The isometric mid-thigh pull (IMTP) is a commonly used method of assessing maximal isometric force production, likely due to its strong association with maximal dynamic force production and performance in numerous athletic tasks [9,10,11,12]. Interestingly, comparing peak force achieved during the CMJ and the IMTP has been suggested to provide insight into the training requirements of athletes by calculating the dynamic strength index (DSI) (CMJ PF/IMTP PF) [13,14,15,16,17,18]. 

The DSI, sometimes also referred to as the dynamic strength deficit, has been suggested to provide insight into an athlete’s training status and potential performance deficit, based on the ratio between the PF during the CMJ and the IMTP [15,16]. For lower body DSI, it has been suggested that a ratio of <0.60 indicates that the athlete should focus on expressing higher forces during ballistic tasks because such a ratio demonstrates that the athlete can express <60% of their maximum isometric force during a ballistic task [15]. In contrast, a ratio of >0.80 is suggested to indicate that the athlete should focus on enhancing maximum force generation as they are capable of expressing >80% of their maximum isometric force during a ballistic task [15]. Interestingly, to date, no data illustrating the changes in DSI in response to different training stimuli have been reported by researchers. 

The aims of this investigation were to determine the effects of a four-week period of in-season strength training on DSI, the components of DSI (namely, CMJ PF and IMTP PF), jump height, and TTT during the CMJ. It was hypothesized that DSI would decrease in response to a training stimulus that focusses on maximal force production, due to an increase in IMTP PF and minimal change in CMJ PF, with any increase in CMJ height a result of an increased TTT. A secondary aim was to determine if athletes with high or low DSI ratios responded differently to a four-week period of in-season strength training, based on the changes in the aforementioned variables. It was hypothesized that both the low and high DSI groups would demonstrate a decrease in DSI as a result of an increase in IMTP PF, in line with previous suggestions [15].

## 2. Materials and Methods

To determine the effect of a period of strength training on DSI and the components of this ratio, a repeated-measures within- and between-subject design was utilized, with subjects assessed at baseline and five days after the final training session of the four-week mesocycle. The between-subject approach was used to determine if any differences in changes in dependent variables were observed between high and low-DSI groups, with an a priori power calculation determining that a minimum of 7 subjects per group was required for a statistical power of ≥0.80 at an a priori alpha level of *p* < 0.05.

Male (*n* = 14, age = 21.9 ± 2.1 years; height = 1.78 ± 0.09 m; mass = 75.8 ± 7.1 kg) and female (*n* = 10, age = 19.1 ± 1.2 years; height = 1.61 ± 0.07 m; mass = 57.3 ± 7.8 kg) collegiate athletes from a variety of sports (rugby league, rowing, field hockey, and bicycle motocross [BMX]) volunteered to participate in this investigation. All subjects provided written informed consent or parental assent as required, and the study was approved by the institutional review board (HSCR16/36).

Prior to testing, subjects performed a non-fatiguing standardized warm up consisting of 20 body weight squats, ten forward and ten reverse lunges, and five submaximal CMJs, with a one-minute rest between exercises, in line with their normal warm-up procedures Further familiarization and three submaximal warm-up trials were performed prior to the maximal IMTP. After completion of the warm up, subjects performed the CMJ and IMTP trials as described below. All subjects were familiar with all testing procedures, as these were included in their usual monitoring procedures. 

CMJ and IMTP performances were assessed with subjects standing on a Kistler force platform—sampling at 1000 Hz—with data collected via Bioware 5.11 software (type 9286AA, Kistler Instruments Inc., Amherst, NY, USA). For the CMJ, subjects stood still for the initial one second of data collection [19,20] to enable the determination of body weight (vertical force averaged over one second). Each subject performed three maximal effort CMJ trials, with a one-minute rest between trials. Subjects were instructed to perform the jumps as fast and as high as possible whilst keeping their hands on their hips. Raw unfiltered force-time data, from the CMJ and IMTP, were used for subsequent analysis [21,22]. 

For the IMTP, the procedures previously described by Haff et al. [23,24] were used, with each subject adopting the posture that they would use for the start of the second pull phase of the clean, resulting in knee and hip angles of 140.1 ± 3.6° and 148.8 ± 4.1°, respectively. All subjects were familiar with this posture through previous performance of the IMTP and the performance of weightlifting exercises and their derivatives within their prior training. Individual joint angles were measured using a hand-held goniometer and recorded and standardized between testing sessions. A cold rolled steel bar was positioned at a height, which replicated the start of the second pull phase of the clean, with the bar fixed above the force platform to accommodate different statures of the individual subjects. Subjects stood on the force platform with their hands strapped to the bar [9], then performed two warm-up trials, separated by one minute of rest, at 50% and 75% of the participant’s perceived maximum effort. 

Once body position was stabilized (verified by watching the participant and force-time record), the participants were given a countdown of “3, 2, 1, Pull!” Minimal pre-tension was permitted (<50 N) to ensure there was no slack in the participant’s body prior to initiation of the pull, with the instruction to pull against the bar “as fast and hard as possible” [25] and push the feet down into the force platform; with this instruction previously reported to produce optimal testing results [25]. Each of the three IMTP trials was performed for approximately five seconds, with strong verbal encouragement provided. If PF during all trials did not fall within 250 N of each other, the trial was discounted and repeated after two minutes of rest, in line with previous recommendations [24,26,27]; all subjects achieved three acceptable trials within 3–5 attempts. 

Although raw force-time data contains greater signal noise than smoothed data, it has been suggested in previous work that analyzing raw CMJ and IMTP force-time data is the criterion method [21,22]. Additionally, all other studies that have calculated the DSI have done so using raw CMJ and IMTP force-time data [13,14,15,16,17,18]. Thus, replicating previous methods in the present study allowed for fairer comparisons of DSI values (and its constituent parts) to these published sources. Raw, unfiltered force-time data were, therefore, exported for subsequent analysis in line with previous recommendations [21,22].

### 2.1. Data Analysis

Force-time data for the CMJ and IMTP were analyzed in Microsoft Excel (Excel 2016, Microsoft, Washington, DC, USA). Jump height was calculated from velocity of center of mass at take-off [20]. Center of mass velocity was determined by dividing vertical force data (minus body weight) by body mass and then integrating the product using the trapezoid rule. The start of the CMJ was identified in line with current recommendations, of 5 standard deviations of bodyweight, during the period of quiet standing -30 ms [19]. Take-off was identified as the point at which vertical force decreased below five times the standard deviation of the force during the flight phase (residual force) [7], and, therefore, TTT was calculated as the duration from the start of the CMJ to take-off. The maximum forces recorded from the force-time curve during the IMTP trials and during the propulsion phase of the CMJ (the start of which was identified based on the velocity of center of mass ≥0.01 m·s^−1^ [3]) were reported as the PF and used to calculate DSI (CMJ PF/IMTP PF). The average value of the three trials was used for statistical analyses. 

Between-session reliability of these measures, with similar athletes, has previously been reported from our laboratory, with very high reliability for the CMJ (intraclass correlation coefficient [ICC] ≥ 0.893, percentage coefficient of variation [%CV] ≤ 5.96) and IMTP (ICC = 0.981, %CV ≤ 4.29) [28]. 

### 2.2. Training Intervention

In line with the normal training schedule, subjects performed the training program (Table 1) two days per week (3–4 days apart, depending on their competition schedule) for four weeks. Loads prescribed for the power clean and mid-thigh pull were based on the subjects’ one repetition maximum (1RM) power clean (which was assessed at the end of their previous training cycle) and the loads prescribed for the remaining exercises were based on predicted 1RM loads based on each subject’s previous 5RM performances, which was determined at the end of their previous phase of training. The volume load during the second session was reduced, via a reduction in the number of exercises performed, as this was the session closest to the subjects’ day of competition. All training sessions were supervised by at least one of the authors, who are qualified strength and conditioning coaches, to ensure consistency of performance.

### 2.3. Statistical Analyses

Normality of all data was determined via Shapiro-Wilk’s test of normality, with all variables being normally distributed. Within-session reliability was determined using two-way random effects model ICC and 95% confidence intervals, and interpreted as low (<0.30), moderate (0.30–0.49), high (0.50–0.69), very high (0.70–0.89), nearly perfect (0.90–0.99), and perfect (1.0) [29]. Percentage coefficient of variation (%CV) was also calculated to determine the within-session variability, with <10% classified as acceptable [30].

Paired samples *t*-tests and Cohen’s *d* effect sizes were calculated to determine if the intervention resulted in any significant or meaningful differences. An a priori alpha level was set at *p* ≤ 0.05, and effect sizes were interpreted as trivial (≤0.19), small (0.20–0.59), moderate (0.60–1.19), large (1.20–1.99), and very large (≥2.0) [31]. All statistical analyses were performed using Statistical Package for the Social Science ([SPSS], Version 23. IBM, New York, NY, USA). 

To identify if subjects with high and low DSI ratios demonstrated different adaptive responses to the four-week period of in-season strength training, the subjects were divided into groups based on the eight highest DSI (*n* = 8; 0.85 ± 0.05 N·N^−1^, range 0.80–0.92 N·N^−1^) and eight lowest DSI values (*n* = 8; 0.56 ± 0.05 N·N^−1^, range 0.46–0.62 N·N^−1^); this division resulted in a significant and very large difference (*p* < 0.001, *d* = 5.80) in DSI between groups. To determine if there were any significant or meaningful differences in the changes in dependent variables between groups, independent *t*-tests and Cohen’s d effect sizes were calculated, with the latter interpreted as described above. 

## 3. Results

Within-session reliability was nearly perfect for all variables (ICC ≥ 0.970) and had very low variability (%CV ≤ 4.69) (Table 2).

Individual changes in CMJ PF, IMTP PF, and DSI are presented in Figure 1. There was a significant yet small (*p* < 0.001, *d* = 0.29) increase in CMJ height post-training (0.34 ± 0.07 m) compared to pre-training (0.32 ± 0.07 m). In contrast, there was no significant or meaningful change (*p* = 0.310, *d* = 0.08) in CMJ PF pre- (1724 ± 337 N) to post- (1754 ± 355 N) training (Figure 2a) and a non-significant yet small increase (*p* = 0.223, *d* = 0.23) in CMJ time to take-off pre- (0.710 ± 0.085 s) to post- (0.731 ± 0.100 s) training. The IMTP PF showed a significant yet small (*p* < 0.001, *d* = 0.38) increase post-training (2761 ± 651 N) compared to pre-training (2509 ± 663 N) (Figure 2b). The combined changes in CMJ and IMTP PF resulted in a significant but small (*p* = 0.009, *d* = 0.50) decrease in DSI post-training (0.71 ± 0.13 N·N^−1^) compared to pre-training (0.65 ± 0.11 N·N^−1^) (Figure 2c).

Individual changes in CMJ PF, IMTP PF, and DSI within the high and low-DSI groups can be seen in Figure 3. There was a non-significant but small (9.56%; *p* = 0.055, *d* = 0.33) increase in CMJ height post-training (0.33 ± 0.05 m) compared to pre-training (0.31 ± 0.07 m) in the high DSI group, whereas the low DSI group demonstrated a significant yet small (6.76%; *p* = 0.007, *d* = 0.23) increase in CMJ height post-training (0.38 ± 0.08 m) compared to pre-training (0.36 ± 0.09 m). There were no significant or meaningful differences (*p* > 0.05, *d* = 0.01) in the change in CMJ height between groups. There was no significant or meaningful change (*p* > 0.05, *d* ≤ 0.36) in CMJ TTT pre- (0.743 ± 0.079 s, 0.679 ± 0.097 s; 4.25%) to post- (0.775 ± 0.098 s, 0.691 ± 0.109 s; 2.18%) training, for the low- and high-DSI groups, respectively. These changes resulted in no significant yet small differences (*p* > 0.05, *d* = 0.31) in the change in TTT between groups. Similarly, there was no significant or meaningful change (*p* > 0.05, *d* ≤ 0.15) in CMJ PF pre- (1698 ± 403 N, 1763 ± 437 N; 3.88%) to post- (1777 ± 328 N, 1779 ± 382 N; 0.09%) training for the low- or high-DSI groups, respectively. These changes resulted in no significant, yet small, differences (*p* > 0.05, *d* = 0.39) in the change in CMJ PF between groups (Figure 4a). The low DSI group showed no significant or meaningful (4.84%; *p* = 0.309, *d* = 0.17) change in IMTP PF pre- to post-training (3066 ± 701 N, 3193 ± 751 N, respectively), whereas the high DSI group demonstrated significant and moderate (15.63%; *p* = 0.004, *d* = 0.73) increases in IMTP PF pre- to post-training (2095 ± 388 N, 2431 ± 524 N, respectively). As a result, there was a non-significant but moderate difference (*p* > 0.05, *d* = 0.74) in the increase in IMTP PF between groups (Figure 4b). The combined changes in CMJ PF and IMTP PF resulted in no significant or meaningful (0.02%; *p* = 1.000, *d* = 0.00) change in DSI pre- to post-training (0.56 ± 0.05 N·N^−1^, 0.56 ± 0.09 N·N^−1^, respectively) for the low DSI group, whereas the high DSI group demonstrated significant and large decreases (12.63% *p* = 0.034, *d* = 1.29) in DSI pre- to post-training (0.85 ± 0.05 N·N^−1^, 0.74 ± 0.11 N·N^−1^, respectively). As a result, there was a significant and moderate difference (*p* = 0.046, *d* = 1.09) in the change in DSI between groups (Figure 4c). 

## 4. Discussion

The results of this study demonstrate that DSI decreases in response to a 4-week period of strength training, as a result of an increase in IMTP PF, with no change in CMJ PF, as hypothesized and in line with previous suggestions [15]. Interestingly and in contrast to our hypotheses, the low DSI group demonstrated no meaningful changes in any of the dependent variables. In contrast, the high DSI group exhibited a large decrease in DSI as a result of a moderate increase in IMTP PF, a result that was in line with our hypotheses. As a result, the high DSI group exhibited a meaningful decrease in DSI compared to the low DSI group. 

As CMJ PF did not demonstrate a meaningful change (~2%) (Figure 2a), the small increase (~7%) in CMJ height post-training was likely due to the small increase (~3%) in TTT post-training, resulting in an increase in net impulse and, therefore, take-off velocity, based on the group means. There was also a small change in CMJ height for the high (~9%) and low (~7%) DSI groups, and while CMJ PF (0% and 4%, respectively) and TTT (4% and 2%, respectively) did increase slightly, these changes were neither significant nor meaningful. It is therefore assumed that the slight increases in PF (Figure 4a) and TTT combined to result in a small increase in net impulse that would have increased the take-off velocity responsible for the small increase in jump height. Individual variation in the changes pre- to post-training can be seen in Figure 1, with more varied responses when divided into low and high DSI groups (Figure 3).

There was a small increase (~11%) in IMTP PF pre- to post-training for the combined data (Figure 2b). Interestingly, the low DSI group showed no meaningful change (~5%) in IMTP PF pre- to post-training, whereas the high DSI group demonstrated a moderate increase (~16%) in IMTP PF pre- to post-training (Figure 4b). These differences are likely a result of the fact that the high DSI group were notably weaker than the low DSI group, based on IMTP PF, with the findings of previous research demonstrating that weaker athletes benefit more from strength training than stronger athletes [32,33].

For the combined data, the small decrease (~7%) in DSI post-training was primarily a result of the increased PF during the IMTP. Between group comparisons revealed a moderate difference in the change in DSI between groups, as the low DSI group exhibited no change in DSI (0%) since the low DSI group demonstrated only small yet comparable changes in both CMJ PF and IMTP PF. In contrast, the high DSI group demonstrated a large decrease (~13%) in DSI pre- to post-training (Figure 4c) due to a moderate increase in IMTP PF and no change in CMJ PF.

The findings demonstrate the importance of (1) increasing force production in weaker athletes and (2) that DSI does decrease, as expected [15], in response to a period of strength training, although four-weeks was not sufficient to reduce the DSI to <0.60, which is suggested to be the point at which rapid force generation should be emphasized. It is suggested that researchers consider the effects of longer periods of strength training and speed strength/power training on DSI to determine if this results in increases in DSI, as previously suggested [15].

It is important that the ratio is not considered alone and that the components of the ratio are considered when using DSI. It is also important to consider if any changes in force production have changed performance, with CMJ height and TTT being potentially useful indicators of performance changes.

## Figures and Tables

**Figure 1 sports-06-00176-f001:**
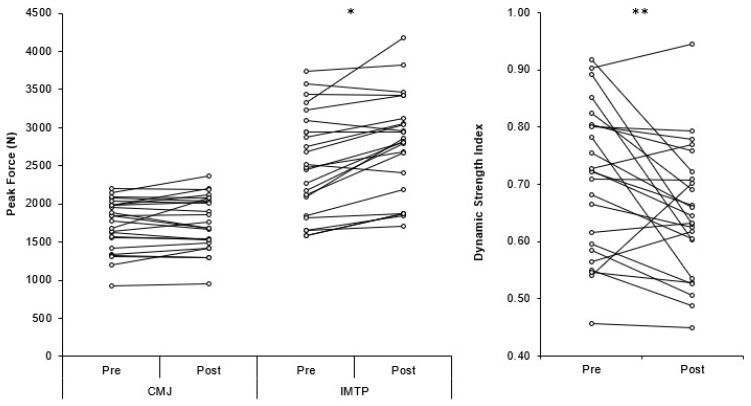
Individual changes in countermovement jump peak force, isometric mid-thigh pull peak force and dynamic strength index. * = significant (*p* < 0.001) difference pre-post training; ** = significant (*p* = 0.009) difference pre-post training

**Figure 2 sports-06-00176-f002:**
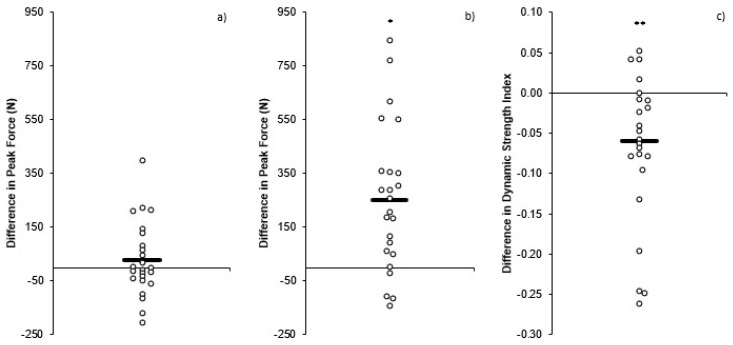
Individual and mean differences pre to post training: (**a**) countermovement jump peak force (**b**) isometric mid-thigh pull peak force (**c**) dynamic strength index. * = significant (*p* < 0.001) difference pre-post training; ** = significant (*p* = 0.009) difference pre-post training

**Figure 3 sports-06-00176-f003:**
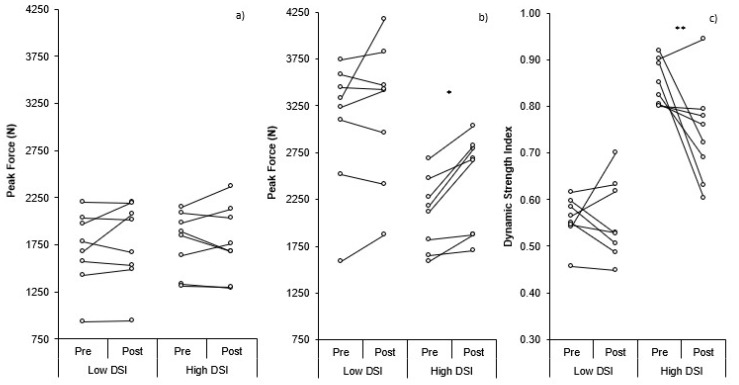
A comparison between low and high dynamic strength index individual changes in: (**a**) countermovement jump peak force, (**b**) isometric mid-thigh pull peak force, and (**c**) dynamic strength index. * = significant (*p* = 0.004) difference pre-post training; ** = significant (*p* = 0.034) difference pre-post training

**Figure 4 sports-06-00176-f004:**
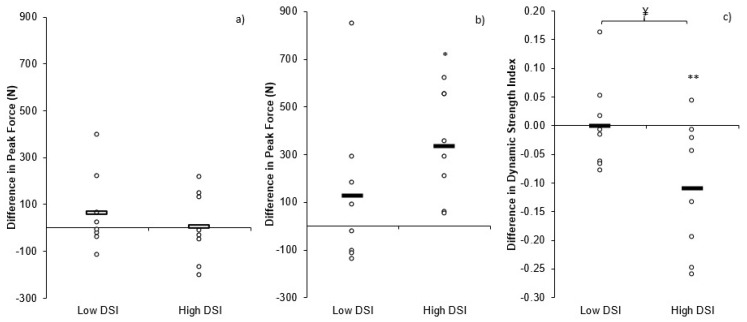
A comparison between low and high dynamic strength index individual and mean differences pre to post training: (**a**) countermovement jump peak force, (**b**) isometric mid-thigh pull peak force, and (**c**) dynamic strength index. * = significant (*p* = 0.004) difference pre-post training; ** = significant (*p* = 0.034) difference pre-post training; ¥ = significant (*p* = 0.046) difference between groups

**Table 1 sports-06-00176-t001:** In season strength training program.

Day 1				
Exercise	Week 1	Week 2	Week 3	Week 4
Power Clean	3 × 3 @ 80%	3 × 3 @ 85%	3 × 3 @ 90%	3 × 3 @ 75%
Push Press	3 × 3 @ 80%	3 × 3 @ 82.5%	3 × 3 @ 85%	3 × 3 @ 75%
Back Squat	3 × 3 @ 85%	3 × 3 @ 87.5%	3 × 3 @ 90%	3 × 3 @ 75%
Nordic Lowers	2 × 3 BW	3 × 3 BW	3 × 3 BW	3 × 3 BW
**Day 2**				
Mid-thigh Pull	3 × 3 @ 80%	3 × 3 @ 82.5%	3 × 3 @ 85%	3 × 3 @ 75%
RDL	3 × 3 @ 80%	3 × 3 @ 82.5%	3 × 3 @ 85%	3 × 3 @ 75%

Sets × Repetitions @ 1RM%, BW = Body Weight; RDL = Romanian Deadlift.

**Table 2 sports-06-00176-t002:** Reliability (intraclass correlation coefficients) and variability (coefficient of variation) of independent variables.

Independent Variable	ICC (95% CI)	%CV
CMJ Height	0.986 (0.972–0.993)	3.89
CMJ Peak Force	0.993 (0.986–0.997)	3.14
CMJ Time to Take-off	0.978 (0.943–0.983)	4.53
IMTP Peak Force	0.994 (0.989–0.997)	3.58
Dynamic Strength Index	0.970 (0.940–0.986)	4.69

CMJ = Countermovement Jump; IMTP = Isometric Mid-Thigh Pull; ICC = Intraclass Correlation Coefficient; CI = Confidence Interval; %CV = Percentage Coefficient of Variation.

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
