# Peer review of "Changes in Dynamic Strength Index in Response to Strength Training"

_sports, 2018, doi:10.3390/sports6040176_

Reviewer 1 Report

Manuscript Title: Changes in dynamic strength index in response to strength training.

 The article is on an interesting topic and would benefit the community. However, there are concerns with the description of the methodology and statistical analysis, and the discussion must be reworked before the paper is suitable for publication.

ABSTRACT:

 Please include a short description of how data were analyzed.

INTRODUCTION:

Line 43: Readability comment: Suggest removing “probably the most extensively used” and replacing is “a commonly used” since the word probably seems to water-down the statement.

The remainder of the introduction is well-written and clear from a reader’s perspective.

METHODS:

Lines 68-70: It is suggested that the authors remove or revise this sentence. In the end of the introduction, the authors describe two aims: 1) to determine the effects of 4-weeks of training on DSI and its critical components, and 2) determine if groups of athletes with low and high DSI, respectively, respond differently to training. This suggests a within-between repeated measures design is necessary, which is not described in the original text. The statistical design section should highlight the within and between nature of the design. Also, did the sample selected provide adequate a priori power so that you are confident the results you obtained are appropriate and that statistical error did not confound the data?

Line 71: Please provide the specific number of men and women and their average age, mass, and height). Also, “weight” is provided in units of kg, which is actually representative of mass. Please change “weight” to “mass” or convert the value to weight units (Newtons).

Lines 75-77: Approximately how long did the warm up last? How many repetitions and sets of each exercise were completed during the warm up? This would help readers determine whether the warm up was sufficient or over- or under-exerting.

Line 88: It would benefit here to add whether the participants were adequately familiar with the phases of the clean exercise and could appropriately create the body orientations of the 2nd pull (i.e., did the knee and hip angular positions match those of their 2nd pull position?).

Line 90-91: How were the joint angles recorded and calculated?

Lines 102-103: Approximately how many trials were needed on average to obtain 3 trials within 250 N of each other? This should be included to help readers determine that fatigue/tiredness did not occur from the isometric pulls.

Lines 107-108: This statement is redundant from above (Lines 86-87).

Line 109 (Data Analysis Section): Based on this section and the statements in lines 86-87 and 107-108, the data were analyzed raw. A single reference is provided to support this method for the both isometric pull and CMJ, but the reviewer is not convinced that a single reference for each movement is sufficient evidence for the assessment of raw data. Particularly in the reference for CMJ, the information relative to the selection of takeoff and the duration of standing used to obtain an appropriate impulse are not in question. However, it is said in the reference that filtered assessments of jump height underestimate CMJ displacement (paraphrasing), assuming that the estimated jump height from raw data is truly representative of the height achieved by the center of mass. However, there is no acknowledgement for the fact that signal noise is a unavoidable consequence of recorded signals and that noise is not a representation of the data being recorded. In addition, it is not clearly described why it is assumed that displacement estimated from raw data is the actual jump height. Finally, no comparisons to kinematics-based assessments of CMJ displacement were included. As such, the authors are encouraged to more thoroughly describe the choice to use raw data in this assessment to 1) provide rationale beyond simply referencing a single study and 2) acknowledge the potential benefits and limitations of the selected method.

Lines 126-133: How was the 1RM power clean determined, and how much time was allotted before the isometric pull and CMJ tests?

Lines 147-151: Based on my comment previously on the within-between design, the statistical analysis section does not follow such an approach. Instead, a within subjects pre-post design using paired t-tests was performed, but the between-subjects design was not the same as the test the reviewer presumed would be used when reading the introduction. The authors are again encouraged to revise the text to more clearly describe this secondary analysis (how were the groups compared statistically [i.e., the magnitude of change of the group responses, low vs high DSI pre-test data and low vs high DSI post-test data, etc.]).

RESULTS:

Tables: No comments, tables are clear

Figures: The authors are commended for presenting individual data in the figures. Still, it would be nice to have markers for statistical significance included in the figures to present which parameters experienced a significant or meaningful (effect size) change pre to post training. Also, it would benefit the figures is a line for the mean value were presented using a dashed line (or similar).

DISCUSSION:

General comment: The discussion is quite short in comparison to the other sections. It is suggested that the authors expand the discussion from reiterating the information from the results (as written) to a synthesis of the results so that researchers and practitioner can apply the knowledge gained from the study. It is suggested that the authors remove the majority of this section, begin the discussion with the information described in lines 215-224, and continue the discussion as appropriate.

Author Response

We would like to thank the reviewer for taking the time to read this manuscript and for providing some excellent feedback which has helped us to enhance the manuscript.

ABSTRACT:

Please include a short description of how data were analyzed.

Response: We have now added some additional detail in light of this suggestion and the suggestions below.

INTRODUCTION:

Line 43: Readability comment: Suggest removing “probably the most extensively used” and replacing is “a commonly used” since the word probably seems to water-down the statement.

The remainder of the introduction is well-written and clear from a reader’s perspective.

Response: Thank you for the positive feedback. We have amended the sentence on line 43 as suggested.

METHODS:

Lines 68-70: It is suggested that the authors remove or revise this sentence. In the end of the introduction, the authors describe two aims: 1) to determine the effects of 4-weeks of training on DSI and its critical components, and 2) determine if groups of athletes with low and high DSI, respectively, respond differently to training. This suggests a within-between repeated measures design is necessary, which is not described in the original text. The statistical design section should highlight the within and between nature of the design. Also, did the sample selected provide adequate a priori power so that you are confident the results you obtained are appropriate and that statistical error did not confound the data?

Response: Apologies for this oversight, we have now amended this in accordance with your suggestions. In addition, we have added the information regarding the a priori power calculation which was performed. ‘To determine the effect of a period of strength training, on DSI and the components of this ratio, a repeated-measures within-between subject design was utilized, with subjects assessed at baseline and five days after the final training session of the four-week mesocycle. The between subject approach was used to determine if any differences in changes in dependent variables were observed between high- and low-DSI groups, with an a priori power calculation determining that a minimum of 7 subjects per group was required for a statistical power of ≥0.80 at an a priori alpha level of p <0.05.’< span="">

Line 71: Please provide the specific number of men and women and their average age, mass, and height). Also, “weight” is provided in units of kg, which is actually representative of mass. Please change “weight” to “mass” or convert the value to weight units (Newtons).

Response: We have now included the individual characteristics for the men and women and replaced weight with mass, as suggested

Lines 75-77: Approximately how long did the warm up last? How many repetitions and sets of each exercise were completed during the warm up? This would help readers determine whether the warm up was sufficient or over- or under-exerting.

Response: We have now included the additional detail, as requested.

Line 88: It would benefit here to add whether the participants were adequately familiar with the phases of the clean exercise and could appropriately create the body orientations of the 2nd pull (i.e., did the knee and hip angular positions match those of their 2nd pull position?).

Response: We have now clarified that the subjects were familiar with the IMTP and the adoption of these postures via the use of weightlifting exercises and their derivatives within their prior training regimes. It was, however, not practical to assess the subjects joint angles during the second pull of the clean and this has only ever been conducted in two studies which incorporated the IMTP.

Line 90-91: How were the joint angles recorded and calculated?

Response: We have now added that this was measured using a hand held goniometer.

Lines 102-103: Approximately how many trials were needed on average to obtain 3 trials within 250 N of each other? This should be included to help readers determine that fatigue/tiredness did not occur from the isometric pulls.

Response: We have now added that this was achieved within 3-5 attempts.

Lines 107-108: This statement is redundant from above (Lines 86-87).

Response: We have amended the text here to avoid unnecessary repetition.

Line 109 (Data Analysis Section): Based on this section and the statements in lines 86-87 and

107-108, the data were analyzed raw. A single reference is provided to support this method for the both isometric pull and CMJ, but the reviewer is not convinced that a single reference for each movement is sufficient evidence for the assessment of raw data. Particularly in the reference for CMJ, the information relative to the selection of takeoff and the duration of standing used to obtain an appropriate impulse are not in question. However, it is said in the reference that filtered assessments of jump height underestimate CMJ displacement (paraphrasing), assuming that the estimated jump height from raw data is truly representative of the height achieved by the center of mass. However, there is no acknowledgement for the fact that signal noise is a unavoidable consequence of recorded signals and that noise is not a representation of the data being recorded. In addition, it is not clearly described why it is assumed that displacement estimated from raw data is the actual jump height. Finally, no comparisons to kinematics-based assessments of CMJ displacement were included. As such, the authors are encouraged to more thoroughly describe the choice to use raw data in this assessment to 1) provide rationale beyond simply referencing a single study and 2) acknowledge the potential benefits and limitations of the selected method.

Response: We have now added the following paragraph to the methods section:

“Although raw force-time data contains greater signal noise than smoothed data, it has been suggested in previous work that analyzing raw CMJ and IMTP force-time data is the criterion method (21, 22). Additionally, all other studies that have calculated the DSI have done so using raw CMJ and IMTP force-time data (13-18), thus replicating previous methods in the present study allowed for fairer comparisons of DSI values (and its constituent parts) to these published sources. Raw, unfiltered force-time data were, therefore, exported for subsequent analysis in line with previous recommendations (21, 22).”

Lines 126-133: How was the 1RM power clean determined, and how much time was allotted before the isometric pull and CMJ tests?

Response: We have now clarified that the 1RM power clean was assessed at the end of the previous phase of training.

Lines 147-151: Based on my comment previously on the within-between design, the statistical analysis section does not follow such an approach. Instead, a within subjects pre-post design using paired t-tests was performed, but the between-subjects design was not the same as the test the reviewer presumed would be used when reading the introduction. The authors are again encouraged to revise the text to more clearly describe this secondary analysis (how were the groups compared statistically [i.e., the magnitude of change of the group responses, low vs high DSI pre-test data and low vs high DSI post-test data, etc.]).

Response: Thank you for your suggestion, we feel that the changes have now improved the manuscript. In line with the data that is presented in Figure 4, we have now also compared the change in CMJ PF, IMTP PF and DSI between groups, using independent t-Tests and Cohen’s d effect sizes (now included this in the statistical analyses description) and described these results in text.

RESULTS:

Tables: No comments, tables are clear

Response: Thank you

Figures: The authors are commended for presenting individual data in the figures. Still, it would be nice to have markers for statistical significance included in the figures to present which parameters experienced a significant or meaningful (effect size) change pre to post training. Also, it would benefit the figures is a line for the mean value were presented using a dashed line (or similar).

Response: We have now added the markers to highlight the significant differences, both within and between groups, based on the significance levels.

We tried including a line for the mean, but this makes the figures look messy, due to the number of lines, and therefore we have left this out at present.

DISCUSSION:

General comment: The discussion is quite short in comparison to the other sections. It is suggested that the authors expand the discussion from reiterating the information from the results (as written) to a synthesis of the results so that researchers and practitioner can apply the knowledge gained from the study. It is suggested that the authors remove the majority of this section, begin the discussion with the information described in lines 215-224, and continue the discussion as appropriate.

Response: We would like to thank the author for their suggestion, however, we have amended the discussion considering the abovementioned changes to the manuscript and the other reviewer and feel that follows a logical flow of results and synthesis of the interactions between the dependent variables, as and when appropriate. It would appear out of place to start the discussion with the information in lines 215-224 (now 253-262), as this would appear out of context without the preceding information contained within the discussion.

 Reviewer 2 Report

General comments

This study examined the influence of strength training on the dynamic strength index. The study is well designed and the manuscript is well constructed and written. Overall, I enjoyed reading this manuscript. I only have a few comments.

Specific comments

P1, L23: “IMTP” and “CMJ” should be spelled out within the abstract.

P3, L131: I could not understand when is the second session. There is no small volume load based on Table 1.

P5, L170: “0.031” should be 0.31.

P7, L207 to 208: “high DSI group were notably weaker than the low DSI group” is based on the value of IMTP PF. And, the CMJ PF is comparable between the groups. Please revise this sentence to avoid misunderstanding of the readers.

Author Response

General comments

This study examined the influence of strength training on the dynamic strength index. The study is well designed and the manuscript is well constructed and written. Overall, I enjoyed reading this manuscript. I only have a few comments.

 Response: Thank you for your kind words and positive feedback

Specific comments

P1, L23: “IMTP” and “CMJ” should be spelled out within the abstract.

Response: Apologies for this oversight, this has now been amended as suggested

P3, L131: I could not understand when is the second session. There is no small volume load based on Table 1.

Response: We have now clarified in the manuscript that the sessions were 3-4 days apart, depending on the athletes’ competition schedule. Volume load is not reported, but it can be clearly seen that the volume has been reduced due to a reduction in the number of exercises performed.

P5, L170: “0.031” should be 0.31.

Response: Apologies for this oversight, this has now been amended

P7, L207 to 208: “high DSI group were notably weaker than the low DSI group” is based on the value of IMTP PF. And, the CMJ PF is comparable between the groups. Please revise this sentence to avoid misunderstanding of the readers.

Response: This has now been amended as suggested

Round  2

Reviewer 1 Report

The authors have adequately addressed this reviewer's original comments/critiques. The review has no major issues with the revised manuscript. Well done.